# Drawback of Enforcing Equivariance and its Compensation via the Lens of Expressive Power

**Yuzhu Chen**[*]                                                              *cyzkrau@mail.ustc.edu.cn*
*University of Science and Technology of China*

**Tian Qin**[*]                                                                  *tqin@mail.ustc.edu.cn*
*University of Science and Technology of China*

**Xinmei Tian**                                                              *xinmei@ustc.edu.cn*
*University of Science and Technology of China*

**Fengxiang He**                                                                *F.He@ed.ac.uk*
*University of Edinburgh*

**Dacheng Tao**                                                          *dacheng.tao@ntu.edu.sg*
*Nanyang Technological University*

**Reviewed on OpenReview:** *https://openreview.net/forum?id=z5bJ44Brc4*

## Abstract

Equivariant neural networks encode the intrinsic symmetry of data as an inductive bias, which has achieved impressive performance in wide domains. However, the understanding to their expressive power remains premature. Focusing on 2-layer ReLU networks, this paper investigates the impact of enforcing equivariance constraints on the expressive power. By examining the boundary hyperplanes and the channel vectors, we constructively demonstrate that enforcing equivariance constraints could undermine the expressive power. Naturally, this drawback can be compensated for by enlarging the model size – we further prove upper bounds on the required enlargement for compensation. Surprisingly, we show that the enlarged neural architectures have reduced hypothesis space dimensionality, implying even better generalizability.

## 1 Introduction

In scaling up the capacity of machine learning models, leveraging the intrinsic symmetry in data, if it exists, would be a much more efficient approach than simply scaling up model size; see good examples in point cloud processing (Qi et al., 2017; Li et al., 2018; Fuchs et al., 2020; Chen et al., 2021), visual tasks (Cohen & Welling, 2016; Xu et al., 2023), graph neural networks (Veličković et al., 2017; Kanatsoulis & Ribeiro, 2024), physics and chemistry (Faber et al., 2016; Eismann et al., 2021; Kondor, 2025), reinforcement learning and decision making (Wang et al., 2022; Qin et al., 2022), amongst others. When learning these symmetries, classic methods must be shown numerous transformed (sometimes manually) data examples, such as rotated or translated images, and be enforced to make invariant/equivariant predictions, which is usually computationally expensive and requires vast amounts of data (Bronstein et al., 2021). Equivariant neural networks address these challenges through either new architectures or optimization methods to enforce symmetries (Cohen & Welling, 2016), in which a useful approach is to enforce each layer of the network to be equivariant.

Despite the impressive performance, it remains unclear whether enforcing equivariance comes at the cost of reduced expressive power. This paper investigates this trade-off by analyzing whether restricting the

---

[*]Both authors contributed equally.

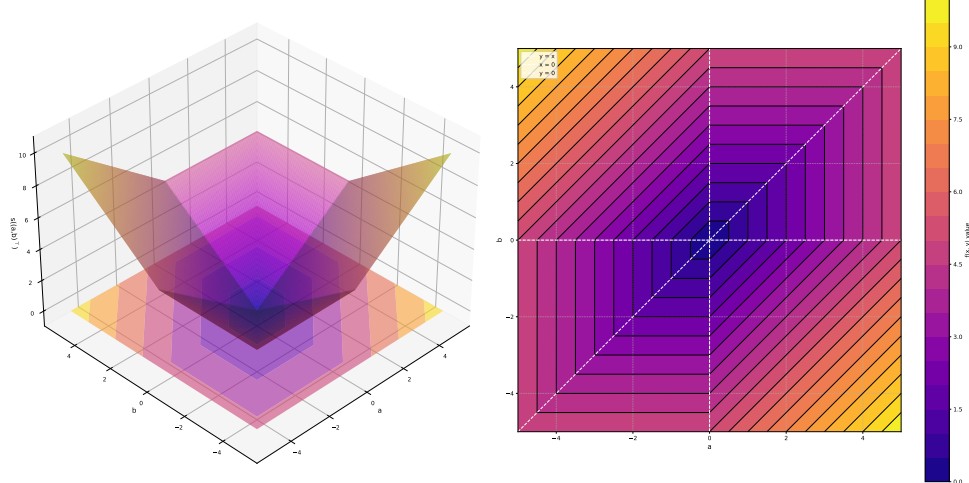

Figure 1: An equivariant function that satisfies $s((a,b)^\top) = s((b,a)^\top)$ and $s((a,b)^\top) = s((-a,-b)^\top)$. The left subfigure is a 3D plot, while the right subfigure is a 2D Contour map.

hypothesis space to equivariant models increases the expected risk when approximating an equivariant target function $s(x)$. Specifically, we examine whether the following inequalities hold:

$$\inf_{\theta \in \Theta} \mathbb{E}_{x \sim P}[\|F_\theta(x) - s(x)\|_2^2] < \inf_{\theta \in \Theta \cap \mathrm{GENs}} \mathbb{E}_{x \sim P}[\|F_\theta(x) - s(x)\|_2^2]$$

$$\text{and} \quad \inf_{\theta \in \Theta \cap \mathrm{GENs}} \mathbb{E}_{x \sim P}[\|F_\theta(x) - s(x)\|_2^2] < \inf_{\theta \in \Theta \cap \mathrm{LENs}} \mathbb{E}_{x \sim P}[\|F_\theta(x) - s(x)\|_2^2].$$

Here, $\Theta$ is the set of network parameters (usually $L$ matrices) and $F_\theta$ is the neural network with parameter $\theta$. We consider two types of equivariant networks: GENs (General Equivariant Networks), where the global mapping $x \to F_\theta(x)$ is required to be equivariant, and LENs (Layer-wise Equivariant Networks), a stricter subset where equivariance is enforced at every layer of the network. In practice, constructing LENs is a common approach to achieving GENs (He et al., 2021).

**Boundary hyperplanes and channel vectors.** We first investigate how these equivariance constraints (imposed at the level of input-output mapping) fundamentally restrict the architectural structure of neural networks. For two-layer ReLU networks, we focus on the geometric properties of the boundary hyperplanes and channel vectors of equivariant networks. We prove that GEN requires boundary hyperplanes to be symmetric. While for LENs, we demonstrate a constructive property: any LEN can be reformulated into another LEN that maintains the same model size but possesses a symmetric set of channel vectors. These geometric rigidities serve as the foundation for our subsequent analysis, enabling us to quantify how much model size is enough for approximating a target function.

**Drawback of enforcing equivariance.** Following the insight of the above analysis, we demonstrate a scenario where equivariance strictly hurts the expressive power. Specifically, we construct an invariant target function $s$ on $\mathbb{R}^2$ and establish two key findings: (1) for two-layer ReLU networks with a single neuron, general networks (GNs) achieve a strictly lower expected error compared to general equivariant networks (GENs), and (2) for two-layer ReLU networks that can arbitrarily approximate the target function, layer-wise equivariant networks (LENs) require a strictly larger model size than GENs.

**Compensations via enlarging model size.** Inversely, we prove that an increase in model size can fix the compromised expressive power. When the target function is invariant, and the data distribution is symmetric, both in terms of group $G$, enlarging the model size of any equivariant network by $|G|$ times can compensate for the expected loss over a 2-layer neural network. Moreover, a doubled model size allows LENs to represent all invariant functions of GENs. We further prove that the constructed LEN with $|G|$ times

enlarged model size surprisingly has a less complex hypothesis space, implying even improved generalizability. Together with the comparable expressive power, this result renders LENs a competitive solution.

To the best of our knowledge, this paper provides the first in-depth theoretical examination of the expressive power of layer-wise equivariant neural networks, demonstrating that enforcing equivariance has drawback in expressive power. We also show that an appropriately larger model size can compensate for the loss in expressive power and potentially enhance model generalisability, underlining the advantages of incorporating layer-wise equivariance into model architectures.

## 2 Related Works

**Enforcing equivariance.** Typical approaches for obtaining an equivariant neural network are (1) designing equivariant architectures (Cohen & Welling, 2017; Satorras et al., 2021; Trang et al., 2024) and (2) introducing equivariance constraints into the optimization (Winter et al., 2022; Tang, 2022; Pertigkiozoglou et al., 2024).

**Equivariant neural architectures.** Cohen & Welling (2016); Shin et al. (2016) design group convolutional neural networks in the group spaces as the equivariant variants of the convolutional neural networks, which is extended to the homogeneous spaces by Cohen et al. (2019). Similar approaches have also been employed in the design of other popular equivariant networks, such as graph neural networks (Klicpera et al., 2020; Aykent & Xia, 2025), transformers (Hutchinson et al., 2021; Islam et al., 2025), and diffusion models (Hoogeboom et al., 2022; Wan et al., 2025). Hao et al. (2025) leverage permutation equivariance in vision transformer for producing "human-imperceptible, machine-recognizable" images.

**Symmetry-aware optimization.** In this stream, equivariance constraints are introduced in optimization, as either regularisers or constraints, to ensure the equivariance, usually termed as steerable neural networks (Cohen & Welling, 2017). Other good examples are about enforcing the symmetry group $S_N$ in steerable graph networks (Maron et al., 2018), the rotation and Euclidean groups $\{C_N, E_2\}$ in steerable convolutional neural networks (Weiler et al., 2018; Weiler & Cesa, 2019), and Lie symmetry in differential equations (Akhound-Sadegh et al., 2023; Jiang et al., 2025). A major obstacle is the high computational costs to compute the equivariant basis. Finzi et al. (2021) address this issues with an algorithm with polynomial computational complexity on the sum of the number of discrete generators and the dimension of the group, which divides the problem into some independent subproblems and adopts Krylov method to compute nullspaces.

**Theoretical analysis.** Theoretical studies have shown that equivariant neural networks have significant advantages, in terms of generalizability, convergence, and approximation. Sannai et al. (2021) prove improved generalization error bounds of equivariant models. Qin et al. (2022) leverage orbit averaging to construct a projection mapping from the original hypothesis space to the permutation-equivalent counterpart, underpinning reduced hypothesis complexity and better generalizability. Lawrence et al. (2022) prove that linear group convolutional neural networks trained by gradient descent for binary classification converge to solutions with low-rank Fourier matrix coefficients based on the results via implicit bias. Zaheer et al. (2017); Maron et al. (2019); Yarotsky (2021); Pacini et al. (2025) prove the universal approximation ability of equivariant models by various approximation techniques, while Ravanbakhsh (2020) utilize group averaging to obtain an equivariant approximator to prove the universal approximation ability. In addition, Elesedy & Zaidi (2021) prove that invariant/equivariant models have a smaller expected loss when the target function is invariant/equivariant. Azizian & marc lelarge (2021) study the expressive power of invariant/equivariant graph neural networks. On the other hand, Petrache & Trivedi (2023) focus on another approach that relaxes the strict constraints to partial and approximate equivariance, and provide bounds on the approximation error and generalization error. In contrast to the aforementioned studies, this paper focuses more on the geometric properties of ReLU networks and their expressive power. We prove that with the same model size, equivariance constraints impair expressive power. For the observed benefits of equivariance, we attribute them to the reduced hypothesis complexity, which holds even if the model size is larger.

## 3 Problem Settings

**General networks (GNs).** GNs refer to general neural networks without any design for realizing the equivariance. In this paper, we focus on two-layer networks, with a hidden layer of width $m$ (also the number of neurons), representing a mapping from $\mathbb{R}^n \to \mathbb{R}^d$. The network is parameterized by $\theta$, including the weight matrix $W^{(1)} \in \mathbb{R}^{m \times n}$ for the hidden layer, and the weight matrix $W^{(2)} \in \mathbb{R}^{d \times m}$ for an output layer. With the popular nonlinear activation ReLU $\sigma$, the GN maps any input $x \in \mathbb{R}^n$ to the output: $F(x) = W^{(2)} \sigma(W^{(1)} x)$. For the brevity, we denote by $\alpha_i^\top$ the $i$-th column of $W^{(1)}$, by $\beta_j$ the $j$-th row of $W^{(2)}$ and $W^{(1)}$, and collect these parameters (with $m$ neurons) into a set $\Theta_m$. In this formulation, the output of a GN can be formulated as follows:

$$F(x) = W^{(2)} \sigma(W^{(1)} x) = \sum_{i=1}^{m} \beta_i \sigma\left(\langle \alpha_i, x \rangle\right), \tag{1}$$

where $\langle \alpha, x \rangle$ is the inner product $\alpha^T x$. We note that the total number of parameters in the GN is $nm + md$. As this count is proportional to the hidden layer size $m$, we use $m$ to characterize the model size.

**General equivariant networks (GENs).** GENs represent a subset of GNs that constrain the output function $F$ to be equivalent with respect to a given group representation. A group representation for a group $G$ is formally defined as a homomorphism $\rho : G \to GL(m)$, where $GL(m)$ is the group of all invertible matrices on $\mathbb{R}^{m \times m}$. A function $F$ is considered equivariant if, for given input and output representations $\rho$ and $\phi$ respectively, it adheres to the condition $F \circ \rho_g = \phi_g \circ F$ for all group elements $g \in G$. A GN whose function $F$ satisfies this property is categorised as a GEN. Specifically, when $\phi = \text{id}$, the function $F$ is called an invariant function and the networks are called invariant networks. In this paper, we assume that group $G$ is finite and focus on invariant functions.

**Layer-wise equivariant networks (LENs).** LENs represent a subset of GNs that constrain each layer to be equivalent with respect to a given group representation. Specifically, given the group representation $\psi$ in the hidden layer, LENs require that for all $g \in G$,

$$\phi_g \circ W^{(2)} = W^{(2)} \circ \psi_g, \quad \psi_g \circ \sigma = \sigma \circ \psi_g, \quad \text{and } \psi_g \circ W^{(1)} = W^{(1)} \circ \rho_g.$$

As each layer is constrained to be equivariant to maintain the symmetry in the data, the whole network is equivariant. Specifically, for every group element $g \in G$, we have

$$F \circ \rho_g = W^{(2)} \circ \sigma \circ W^{(1)} \circ \rho_g = W^{(2)} \circ \sigma \circ \psi_g \circ W^{(1)},$$
$$= W^{(2)} \circ \psi_g \circ \sigma \circ W^{(1)} = \phi_g \circ W^{(2)} \circ \sigma \circ W^{(1)} = \phi_g \circ F.$$

In practice, constructing LENs is a common approach to achieving GENs (He et al., 2021).

**Expressive power.** The expressive power, of GNs, GENs, and LENs is defined as the minimum expected loss for approximating a target function $s$. Naturally, a smaller minimum expected loss means a better expressive power. Since GENs are in a subset of GNs, and LENs have intuitively more constraints, we have the following straightforward results: for any target function $s$, any input distribution $P \in \Delta \mathbb{R}^n$, and any number of neurons $m$,

$$\inf_{\theta \in \Theta_m \cap \text{GNs}} \mathbb{E}_{x \sim P}[\|F_\theta(x) - s(x)\|_2^2] \leq \inf_{\theta \in \Theta_m \cap \text{GENs}} \mathbb{E}_{x \sim P}[\|F_\theta(x) - s(x)\|_2^2] \leq \inf_{\theta \in \Theta_m \cap \text{LENs}} \mathbb{E}_{x \sim P}[\|F_\theta(x) - s(x)\|_2^2],$$

where $\theta$ represents the parameter of a two-layer ReLU network ($\theta = (W^{(2)}, W^{(1)})$), and $F_\theta$ is the forward function given parameter $\theta$, and $\Theta_m$ is the parameter set of $m$-hidden-layer networks. We reuse the notations GNs (also for GENs and LENs) to denote the respective parameter sets that makes $F_\theta$ a GN (also for GEN and LEN). This paper investigates whether equivariance constrains the expressive power of two-layer networks. Specifically, we study whether there exists some equivariant function $s$ and $m$ such that

$$\inf_{\theta \in \Theta_m \cap \text{GNs}} \mathbb{E}_{x \sim P}[\|F_\theta(x) - s(x)\|_2^2] < \inf_{\theta \in \Theta_m \cap \text{GENs}} \mathbb{E}_{x \sim P}[\|F_\theta(x) - s(x)\|_2^2], \text{ and}$$
$$\inf_{\theta \in \Theta_m \cap \text{GENs}} \mathbb{E}_{x \sim P}[\|F_\theta(x) - s(x)\|_2^2] < \inf_{\theta \in \Theta_m \cap \text{LENs}} \mathbb{E}_{x \sim P}[\|F_\theta(x) - s(x)\|_2^2].$$

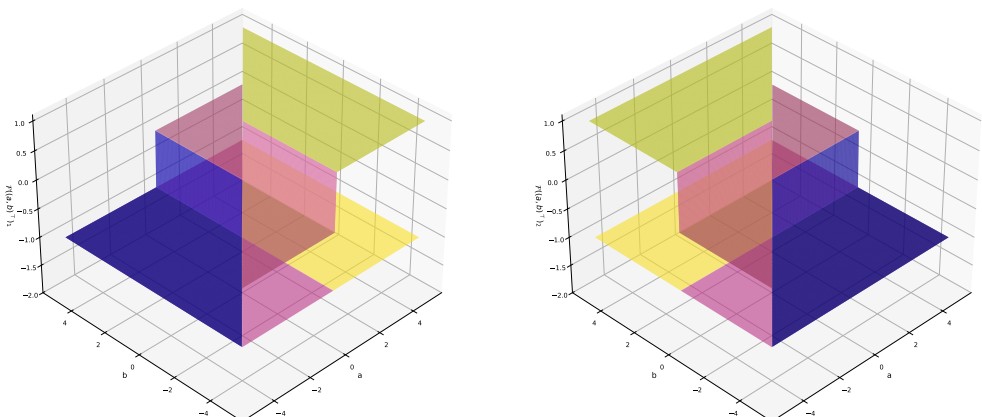

Figure 2: A visualization of the feature function $\mathcal{F}$ of $F(x,y) = \sigma(x) + \sigma(-y) + \sigma(-x+y)$, where the left subfigure is $\mathcal{F}_1$ and the right figure is $\mathcal{F}_2$. As shown, there are two boundary hyperplanes: $x = 0$, $y = 0$, $y = x$.

The first inequality studies whether being equivariant is necessary for being the optimal approximator if the target function is equivariant. The second is on whether enforcing layer-wise equivariance is the optimal way to incorporate equivariance. To ensure a fair comparison, we compare the expressive power of GNs, GENs, and LENs under the same architecture (e.g., the same weight-sharing scheme) and the same model size. We also note that it is reasonable to assume the target function $s$ to be equivariant, because equivariant networks are designed for equivariant tasks.

## 4 Boundary Hyperplanes and Channel Vectors

To establish a clear foundation for our analysis of expressive power, we discuss two tools, symmetry boundary hyperplanes and symmetry channel vectors, in this section.

### 4.1 GENs Imply Symmetric Boundary Hyperplanes

Naturally, ReLU networks are also piece-wise linear, given ReLU functions are piece-wise linear:

$$F(x) = \sum_{i=1}^{m} \beta_i \sigma\left(\langle \alpha_i, x \rangle\right) = \sum_{i:\langle \alpha_i, x \rangle \geq 0}^{m} \beta_i \alpha_i^\top x = \left\langle \sum_{i:\langle \alpha_i, x \rangle \geq 0}^{m} \alpha_i \beta_i^\top, x \right\rangle := \langle \mathcal{F}(x), x \rangle, \tag{2}$$

where $\mathcal{F}(x) = \sum_{i:\langle \alpha_i, x \rangle \geq 0}^{m} \alpha_i \beta_i^\top$ is the feature function of $F$ and is piece-wise constant.

We define the boundary hyperplane as the hyperplane whose feature function values are almost surely different on its two distinct sides. For simplicity, we assume that all hyperplanes contain 0 and have dimension $n - 1$. The formal definition is shown below.

**Definition 4.1** (boundary hyperplane)**.** *Let $F$ be an output function of a two-layer ReLU network and $\mathcal{F}(x)$ be the corresponding feature function. We call a hyperplane $M$ a boundary hyperplane if and only if for all $x \in M$ and $v \notin M$:*

$$\lim_{\delta \to 0^+} \mathcal{F}(x + \delta v) \neq \lim_{\delta \to 0^+} \mathcal{F}(x - \delta v).$$

It is not necessary to define the feature function $\mathcal{F}$ on the boundary hyperplanes. For any given function $F$, we can define it only on the set $S = \{x : F \text{ is linear in } B(x, r), \exists r > 0\}$, where $B(x, r) = \{x + z :$

$\|z\| < r\}$ is the open ball centering at $x$ with radius $r$. The value of $\mathcal{F}(x)$ is determined uniquely by the Riesz representation theorem. The complementary set $S^c$ is the union of all boundary hyperplanes, whose intersection with the hyperplanes $M \pm \delta v$. For all $x \in S$ that satisfies $\langle x, \alpha_i \rangle \neq 0$ for all $i \in [m]$, $\mathcal{F}(x)$ must be $\sum_{i:\langle x, \alpha_i \rangle > 0} \beta_i \alpha_i^T$. Therefore, if the feature function changes when crossing through the same hyperplane, the active set $\{i : \langle x, \alpha_i \rangle > 0\}$ must change and the increment of the feature function is $\sum_{i:\langle \alpha_i, x \rangle = 0} \beta_i \alpha_i^T$, which is then summarized as Theorem 4.2. This theorem can be viewed as an equivalent definition by the channel vectors.

**Theorem 4.2.** *Let $F = \sum_{i=1}^m \beta_i \sigma(\langle \alpha_i, x \rangle)$ be an output function. Then, a hyperplane $M$ is a boundary hyperplane if and only for all $v \notin M$:*

$$\sum_{i:\langle \alpha_i, v \rangle > 0, \langle \alpha_i, M \rangle = 0} \alpha_i \beta_i^T \neq \sum_{i:\langle \alpha_i, v \rangle < 0, \langle \alpha_i, M \rangle = 0} \alpha_i \beta_i^T,$$

*where $\langle \alpha_i, M \rangle = 0$ are defined as $\langle \alpha_i, x \rangle = 0$ for all $x \in M$. Especially, if $M$ is a boundary hyperplane, there exists some channel vector $\alpha$ such that $\langle \alpha, M \rangle = 0$.*

Boundary hyperplane provides a powerful analytical tool for understanding the internal mechanisms of GENs. We prove that for a GEN, the set of its boundary hyperplanes is symmetric with respect to all group representations. Specifically, for any given boundary hyperplane $M$, then for any $g \in G$, the transformed hyperplane $\rho_g M = \{\rho_g x : x \in M\}$ is also a boundary hyperplane of the GEN, because

$$\lim_{\delta \to 0^+} \mathcal{F}(\rho_g x + \delta \rho_g v) = (\rho_g^T)^{-1} \lim_{\delta \to 0^+} \mathcal{F}(x + \delta v) \neq (\rho_g^T)^{-1} \lim_{\delta \to 0^+} \mathcal{F}(x - \delta v) = \lim_{\delta \to 0^+} \mathcal{F}(\rho_g x - \delta \rho_g v),$$

where $x$ is a point on the hyperplane $M$ and $v$ is any vector not in $M$. The inequality holds because the group representation $\rho_g$ is always invertible. We note that $\rho_g M$ is not always another hyperplane. In fact, for the identity $g = e$, $\rho_e M = M$.

This property hold for all group elements $g \in G$, implying the set of all transformed hyperplanes, $\{\rho_g M : g \in G\}$, is a subset of the GEN's total set of boundary hyperplanes. It also holds for all boundary hyperplanes $M$, which means the GEN's total set of boundary hyperplanes is symmetry under $\rho$. We summarize this finding in the following theorem.

**Theorem 4.3.** *For a GEN on group $G$, its boundary hyperplane set $\mathcal{M}$ is closed under group representation $\rho_g$ for all $G$.*

This theorem has direct implications for the expressive power of GENs. When approximating a target function, the network must capture its discontinuous, i.e., boundary hyperplanes. Due to the symmetry requirement, even a single learned hyperplane $M$ generates a whole family of hyperplanes, $\{\rho_g M : g \in G\}$. The size of this set provides a lower bound on the number of boundary hyperplanes the GEN must use to approximate the target function effectively. According to Theorem 4.2, a larger number of required hyperplanes, in turn, necessitates a greater number of neurons in the network.

## 4.2 LENs further Imply Symmetric Channel Vectors

LENs represent a more constrained subset of GENs, enforcing symmetry at the layer level rather than across the network as a whole. Although the symmetry property for boundary hyperplane is a fundamental property of all GENs, the stricter conditions imposed by the LEN structure provide deeper insights. At least, as mentioned in Section 3, LEN needs one admitted group representation and two weight matrices in the corresponding intertwining space.

We first consider the property of admitted group representation, i.e., $\psi \circ \sigma = \sigma \circ \psi$. This commutation property is not trivial and places a significant restriction on the form that the group representation $\psi$ can take. Specifically, for $\sigma = \text{ReLU}$, the following lemma provides a complete characterization of all such admitted representations.

**Lemma 4.4** (Godfrey et al. (2022); Bökman & Kahl (2023)). *For $\sigma = \text{ReLU}$, a group representation $\psi$ is admitted if and only if for all $g \in G$, $\psi_g$ is a generalized permutation matrix with exclusively positive*

*non-zero entries. That is, each row and each column $\psi_g$ must contain exactly one non-zero element, and that element must be positive.*

*Proof.* We first prove that each row of $\psi_g$ has at most one non-zero entry for all $g \in G$. Assume for contradiction that exists $g \in G$ where a row $i$ of $\psi_g$ has at least two non-zero entries, $(\psi_g)_{ij}$ and $(\psi_g)_{ik}$. Without loss of generality, we assume $(\psi_g)_{ij} \leq (\psi_g)_{ik}$. Admitted $\psi$ means that for input vector $x = e_j - e_k$, we have $(\sigma(\psi_g x))_i = (\psi_g \sigma(x))_i = (\psi_g e_j)_i = (\psi_g)_{ij}$. However, on the other side we have $(\sigma(\psi_g x))_i = \sigma((\psi_g)_{ij} - (\psi_g)_{ik}) = 0$, which contradicts our initial assumption that the entries were non-zero. Therefore, each row of $\psi_g$ contains at most one non-zero entry.

Each row has exactly one non-zero entry because $\psi_g$ is invertible, which also means each entry has at least one non-zero row, making $\psi_g$ a generalized permutation matrix. Moreover, when $(\psi_g)_{ij} \neq 0$, we can set $x = e_j$. Then $\sigma(\psi_g x) = \psi_g \sigma(x)$ implies that $\sigma(\psi_{ij}) = \psi_{ij}$. From the definition of ReLU, $(\psi_g)_{ij}$ is positive for all $i$. $\square$

Lemma 4.4 implies that the action of any $\psi_g$ on a vector is a composition of a permutation and a positive scaling of its coordinates. For each $g \in G$, we can identify a unique permutation $P_g$ that maps the coordinate indices. Specifically, $P_g(i)$ is the index of the only non-zero element in row $i$ of matrix $\psi_g$. A special case of an admitted group representation occurs when all non-zero entries of $\psi_g$ is 1. In this scenario, each $\psi_g$ is a standard permutation matrix. We refer to this $\psi$ as a permuting representation.

Following the property of admitted representation, we now examine how it interacts with the other constraints of invariant LENs. The final linear layer, represented by $W^{(2)} = [\beta_1, \cdots, \beta_m]$, is required to satisfy the condition $W^{(2)} \circ \psi = W^{(2)}$ for LENs. This requirement reveals a deeper constraint on $\psi$ and the geometry of $W^{(2)}$. Specifically, we have the following lemma:

**Lemma 4.5.** *For admitted $\psi$ in an invariant LEN and any connected pair of indices $(i, j)$ (exists $g \in G$ making $P_g(i) = j$), we have $\|\beta_j\|\beta_i = \|\beta_i\|\beta_j$. Besides, all $h \in G$ with $P_h(i) = j$ shares the same positive $(\psi_h)_{ij}$.*

*Proof.* Consider a pair $(i, j)$ that makes $\psi_{ij} \neq 0$ for some $g \neq h$. Admitted $\psi$ means that for input $e_j$, we have $\beta_j = W^{(2)} e_j = W^{(2)} \psi e_j = \psi_{ij} W^{(2)} e_i = \psi_{ij} \beta_i$ for all $\psi$, including $\psi_g$ and $\psi_h$. It shows that $\beta_i$ and $\beta_j$ share the same direction, meaning $\|\beta_j\|\beta_i = \|\beta_i\|\beta_j$. It also shows that $\psi_{ij}$ does not change if $\psi$ switches from $\psi_g$ to $\psi_h$, i.e., $(\psi_g)_{ij} = (\psi_h)_{ij}$. $\square$

In light of Lemma 4.5, we say $\beta_i$ and $\beta_j$ are in the same orbit if they share the same direction, i.e., $\|\beta_j\|\beta_i = \|\beta_i\|\beta_j$. The lemma demonstrates that the symmetry constraint requires all weight vectors in the same orbit to lie in the same direction in vector space.

Aside from the constraint on $W^{(2)}$, LENs also impose a condition on $W^{(1)}$ - $W^{(1)}$ should belong to the intertwiner space defined by $\psi$ and $\rho$. Combining it with the constraints on $\psi$ and $W^{(2)}$, we provide the following theorem.

**Theorem 4.6.** *For any $\rho$-invariant LEN $F_\theta$, there exists an equivalent $\rho$-invariant LEN, $F_{\tilde{\theta}}$ of the same size that computes the exact same function ($F_{\tilde{\theta}}(x) = F_\theta(x)$ for all $x$), where the channel vectors set of the rewritten network ($\{\tilde{\alpha}_i\}$) is closed under the action of the transposed group representation $\rho^\top$. That is, for any channel vector $\tilde{\alpha}_i$ and any group element $g \in G$, $\rho_g^\top \tilde{\alpha}_i$ is also a channel vector of $F_{\tilde{\theta}}$.*

The proof is included in Appendix D. The theorem means that we can rewrite any LEN into another LEN of the same size, where the channel vectors of the rewritten LEN exhibit an explicit symmetry in $\rho^\top$. In the view of input-output mapping, the rewritten LEN is equivalent to the given LEN. For the rewritten LEN, the requirement of symmetry channel vectors means that one single channel vector $\tilde{\alpha}_i$ implicitly defines a subset of channel vectors $\{\rho_g^\top \tilde{\alpha}_i : g \in G\}$. The network size is then smaller and bounded to accommodate all these channel vectors.

We note that this constraint is stricter than the one for GENs. A GEN requires the symmetry for boundary hyperplanes while LEN requires that for channel vectors, where one boundary hyperplane has to be

determined by at least one channel vector, but different channel vectors could lead to one hyperplane. For instance, the channel vectors $\alpha$ and $-\alpha$ are distinct vectors but define the exact same boundary hyperplane. A GEN has the flexibility to use scaled versions of the same vector normal, whereas a LEN, in its symmetric form, must treat each parallel vector as a distinct channel vector.

## 5 Drawback of Equivariance in Expressive Power

In this section, we show our results on how enforcing equivariance hurts the expressive power. The proofs are through constructing a target function on $\mathbb{R}^2 \to \mathbb{R}$. When approximating this target function, we show that for neural networks with one neuron, i.e. in parameter set $\Theta_1$, GNs have strictly smaller expected loss than GENs. For 3-neuron-networks, GNs and GENs can fully approximate this target function, while LENs cannot.

**Example 5.1.** *Consider group $G$ consists of 4 elements $e, g, h, gh$ where $g^2 = h^2 = e$ and $g \cdot h = h \cdot g = gh$. One group representation $\rho$ is as follows*

$$\rho_e = \begin{bmatrix} 1 & 0 \\ 0 & 1 \end{bmatrix}, \quad \rho_g = \begin{bmatrix} 0 & 1 \\ 1 & 0 \end{bmatrix}, \quad \rho_h = \begin{bmatrix} -1 & 0 \\ 0 & -1 \end{bmatrix}, \quad and \quad \rho_{gh} = \begin{bmatrix} 0 & -1 \\ -1 & 0 \end{bmatrix}.$$

*Given representation $\rho$, consider target function $s$:*

$$s : \mathbb{R}^2 \to \mathbb{R}, \qquad (a, b)^\top \to \max(|a|, |b|, |a - b|),$$

*which is an invariant target function because*

$$s \circ \rho_g \begin{pmatrix} a \\ b \end{pmatrix} = s \begin{pmatrix} b \\ a \end{pmatrix} = \max(|b|, |a|, |b - a|) = s \begin{pmatrix} a \\ b \end{pmatrix}, \; and$$

$$s \circ \rho_h \begin{pmatrix} a \\ b \end{pmatrix} = s \begin{pmatrix} -a \\ -b \end{pmatrix} = \max(|-a|, |-b|, |b - a|) = s \begin{pmatrix} a \\ b \end{pmatrix}.$$

*We note that $s$ has another form: $s((a, b)^\top) = \sigma(a) + \sigma(-b) + \sigma(b - a)$.*

Figure 1 provides a 3D plot and a 2D contour map to visualize the effects of the imposed symmetry. Figure 2 displays the feature function of the symmetry group element. Building on this example, we demonstrate how the constraints of GENs can limit expressive power and how this limitation is further pronounced in LENs. Specifically, we mainly focus on comparisons of GENs vs. GNs on the 1-neuron setting, and LENs vs GENs on the 3-neuron setting.

**One-neuron networks.** The output function of the neural network becomes $\beta\sigma(\langle \alpha, x \rangle)$. A typical one-neuron network applies the ReLU activation function to each dimension of the input. Formally, when $\alpha = (1, 0)^\top$ and $\beta = 1$, $F((a, b)^\top) = \sigma(a)$ is an approximation and $F$ is a GN with one neuron. Similarly, when $\alpha = (0, 1)^\top$ and $\beta = 1$, $F((a, b)^\top) = \sigma(b)$ is also a GN approximation. Thus, for all input distribution $P$ with $\Pr_{x \sim P}[x^\top = (0, 0)] < 1$, we have

$$\inf_{\theta \in \Theta_1 \cap \text{GNs}} \mathbb{E}_{x \sim P}[\|F_\theta(x) - s(x)\|_2^2] \le \min\left[\mathbb{E}_{x \sim P}\|\sigma(a) - s(x)\|^2, \mathbb{E}_{x \sim P}\|\sigma(b) - s(x)\|^2\right] < \mathbb{E}_{x \sim P}\|s(x)\|^2. \quad (3)$$

For GENs, we have $f(\rho x) = f(x)$. Combining with the definition of feature function $\mathcal{F}$, we have $\langle \mathcal{F}(\rho x), \rho x \rangle = \langle \rho^T \mathcal{F}(\rho x), x \rangle = \langle \mathcal{F}(x), x \rangle$, which implies $\rho^T \circ \mathcal{F} \circ \rho = \mathcal{F}$ for all $\rho$. Let $\rho = \rho_h$, we have for all $x$:

$$\beta\sigma(\langle \alpha, x \rangle) = \mathcal{F}(x) = (\rho_h^T \circ \mathcal{F} \circ \rho_h)(x) = -\beta\sigma(\langle \alpha, -x \rangle),$$

where at least one of $\sigma(\langle \alpha, x \rangle)$ and $\sigma(\langle \alpha, -x \rangle)$ is zero, indicating $\mathcal{F} = 0$ for all $x$. It shows that the only one-neuron GEN is zero, which leads to strictly lower expressive power then GNs. Specifically, we have that for all input distribution $P$ with $\Pr_{x \sim P}[x^\top = (0, 0)] < 1$,

$$\inf_{\theta \in \Theta_1 \cap \text{GENs}} \mathbb{E}_{x \sim P}[\|F_\theta(x) - s(x)\|_2^2] = \mathbb{E}_{x \sim P}\|s(x)\|^2 > \inf_{\theta \in \Theta_1 \cap \text{GNs}} \mathbb{E}_{x \sim P}[\|F_\theta(x) - s(x)\|_2^2]. \quad (4)$$

**Three-neuron networks.** Three-neuron GNs can represent $s$ defined in Example 5.1 because the form $s((a,b)^\top) = \sigma(a) + \sigma(-b) + \sigma(b-a)$ is already a three-neuron network. Due to the equivariance property of $s$, this network is also a GEN. Therefore, three neurons are sufficient for both GNs and GENs to achieve $s$.

However, for LENs, we show that three neurons are not enough. According to Theorem 4.6, for any LEN that represents $s$, there exists a rewritten LEN with symmetric channel vectors of the same size that also perfectly represents $s$. Since $s$ has three boundary hyperplanes: $x = 0$, $y = 0$, and $x - y = 0$, they should also be the boundary hyperplane of the rewritten LEN, which implies the existence of at least three channel vectors: $c_1(1,0)$, $c_2(0,1)$, and $c_3(1,1)$ with $c_1 c_2 c_3 \neq 0$. Since the channel vectors are symmetric, $\rho_h^\top c_1(1,0) = -c_1(1,0)$also forms a channel vector; as do $-c_2(0,1)$ and $-c_3(1,1)$. As a result, the original LEN must contain at least six channel vectors, i.e., LENs require at least six neurons to represent $s$. Finally, we show that six neurons are sufficient, as the following LEN has exact six neurons and represents $s$:

$$F_\theta((a,b)^\top) = \frac{1}{2}[\sigma(a) + \sigma(-a) + \sigma(b) + \sigma(-b) + \sigma(b-a) + \sigma(a-b)].$$

# 6 Compensation to Enforcing Equivariance

In the previous sections, we demonstrated that for a fixed model size, the expressive power of LENs are strictly lower than that of GENs, which are also strictly less expressive than GNs. A natural question arises: can this limitation in expressive power be compensated for by enlarging the model size? This section affirms that enlarging the model size is indeed a viable solution. We will examine this through both realizable and non-realizable cases. Further, we show that, surprisingly, the enlarged networks can have even reduced hypothesis complexity, which leads to better generalizability.

## 6.1 Realizable Case

We first consider the realizable case, where the target function can be perfectly represented by a GN with a finite number of neurons. By definition, $s$ can also be represented by a finite-neuron GEN. When it comes to LEN, the conclusion is not straightforward.

Firstly, we show that with a larger model size, there always exists a LEN that can represent any representable mapping of finite-neuron GENs. To demonstrate this, consider a target GEN-representable function defined as $s(x) = \sum_{i=1}^m \beta_i \sigma(\langle \alpha_i, x \rangle)$. We can construct a LEN, denoted by $F(x)$, that perfectly replicates $s(x)$. The weights for this constructed LEN are defined as follows:

$$W^{(2)} = \frac{1}{|G|}[\underbrace{\beta_1, \ldots, \beta_1}_{|G|}, \ldots, \underbrace{\beta_m, \ldots, \beta_m}_{|G|}], \text{ and}$$

$$W^{(1)} = [\rho_{g_1}^T \alpha_1, \ldots, \rho_{g_{|G|}}^T \alpha_1, \ldots, \rho_{g_1}^T \alpha_m, \ldots, \rho_{g_{|G|}}^T \alpha_m]^T.$$

This construction ensures that the weights adhere to the necessary equivariance conditions. In the first layer, for all $g \in G$, $\rho_{g_i} \langle \alpha_k, \rho_{g_i}(x) \rangle = \langle \rho_{g_i}^\top \rho_{g_i}^\top \alpha_k, x \rangle = \langle \rho_{g_i g_j}^\top \alpha_k, x \rangle$, implying the existence of a permutation $\psi$ that maps the neuron index corresponding to $\rho_{g_i}^\top \alpha_k$ to the index for $\rho_{g_i g_j}^\top \alpha_k$. This $\psi$ is admitted because of Theorem 4.4. In the second layer, we have that the $\beta$ is $\beta$ for all $\rho_h^\top \alpha_i$, indicating its invariance with respect to $\psi$.

This constructed LEN is a rewritten version of $s$ because of its invariance and equivariance; specifically,

$$F(x) = \sum_{i=1}^m \frac{1}{|G|} \beta_i \sum_{g \in G} \sigma\left(\langle \rho_g^\top \alpha_i, x \rangle\right)$$

$$= \frac{1}{|G|} \sum_{g \in G} \sum_{i=1}^m \beta_i \sigma\left(\langle \alpha_i, \rho_g x \rangle\right)$$

$$= \frac{1}{|G|} \sum_{g \in G} s(\rho_g(x)) = s(x),$$

which confirms that our constructed LEN perfectly represents the target function $s$. This result can be extended to handle any equivariant functions and multi-layer architectures, as detailed in Appendix E.

Building on this, we utilize the relationship between the number of boundary hyperplanes and that of channel vectors to study how many neurons are required to let LEN approach GEN. For GENs or LENs, the number of channel vectors is larger than the number of boundary hyperplanes, according to Theorem 4.2. Inversely, for $f \in$ GENs, if there are $k$ boundary hyperplanes of some output function $k$, we can construct some specific $\beta_i = 1$ and $\alpha_i$ for $i \in [k]$ such that $f - \sum_{i=1}^{k} \beta_i \sigma(\langle \alpha_i, x \rangle)$ has no boundary hyperplanes, which then can be represented as a linear function $\langle \alpha_0, x \rangle = \sigma(\langle \alpha_0, x \rangle) - \sigma(\langle -\alpha_0, x \rangle)$. Therefore, for GENs, we can compress neurons to make the number of channel vectors equal to the number of boundary hyperplanes plus two.

As for LENs, we prove the following lemma that deals with all symmetric channel vectors as a whole.

**Lemma 6.1.** *Let $\sum_{i=1}^{\ell} \beta_i \sum_{g \in G} \sigma(\langle \rho_g^T \alpha_i, x \rangle)$ be an output function of LENs with $\alpha_i \neq 0$ for all $i \in [\ell]$. Denote $M_i = \{\rho_g^T \alpha_i / \|\rho_g^T \alpha_i\| : g \in G\}$ as the normalized orbit induced by $\alpha_i$. Then, we have $M_i = M_j$ or $M_i \cap M_j = \phi$ for all $i, j \in [\ell]$. Moreover, if $M_i = M_j$, we have*

$$\sum_{g \in G} \sigma(\langle \rho_g^T \alpha_i, x \rangle) = k_{ij} \sum_{g \in G} \sigma(\langle \rho_g^T \alpha_j, x \rangle),$$

*which enables us to reformulate the output function as $\sum_{i=1}^{\ell'} \tilde{\beta}_i \sum_{g \in G} \sigma(\langle \rho_g^T \tilde{\alpha}_i, x \rangle)$ such that $\tilde{M}_i \cap \tilde{M}_j = \phi$ for all $i, j \in [\ell']$. Furthermore, for each $\tilde{M}_i$, we can then merge all parallel channel vectors, since*

$$\sum_{g \in G} \sigma(\langle \rho_g^T \tilde{\alpha}_i, x \rangle) = |Stab(\tilde{\alpha}_i)| \sum_{g \in G/Stab(\tilde{\alpha}_i)} \sigma(\langle \rho_g^T \tilde{\alpha}_i, x \rangle),$$

*where $Stab(\alpha) = \{g \in G : \rho_g^T \alpha = \alpha\}$ is the stabilizer subgroup with respected to $\alpha$. As a result, in the equivalent form $\sum_{i=1}^{\ell'} \tilde{\beta}_i |Stab(\tilde{\alpha}_i)| \sum_{g \in G/Stab(\tilde{\alpha}_i)} \sigma(\langle \rho_g^T \tilde{\alpha}_i, x \rangle)$, there are not two channel vectors of the same direction.*

According to Lemma 6.1, we can construct LENs whose channel vectors are not parallel. Hence, the number of channel vectors is no more than double the number of boundary hyperplanes plus two. Therefore, the minimum required model size of LENs is approximately no more than double that of GENs, which is summarized in the following theorem with some technical modifications.

**Theorem 6.2.** *LENs require at most double model size to represent all invariant output functions of GENs.*

Theorem 6.2 indicates that enforcing layer-wise equivariance would be competitive among all methods of incorporating equivariance. We note that the conclusion of Theorem 6.2 can alternatively be derived from the work of Agrawal & Ostrowski (2022), where a canonical form of GENs is defined, and doubling of model size makes each layer equivariant. In contrast, we present an independent constructive proof in Appendix G, which utilizes group averaging and the orbit merging of channel vectors. Our proof places greater emphasis on the intrinsic geometric properties of LENs, and this geometric perspective further underpins the analysis of expressive power for LENs in Section 6.2.

## 6.2 Non-realizable Case

We now consider the non-realizable problem. With the same model size, enforcing equivariance may hurt the expressive power. We prove that when the networks have a larger model size, LENs may achieve a comparable and even better expressive power. We start with the following theorem, which shows a comparable expressive power when the model size is enlarged $|G|$ times.

**Theorem 6.3** (Elesedy & Zaidi (2021)). *Let $s$ be any invariant target function. Denote by $\mathcal{Q}f$ the projection of $f$ to the equivariant space, defined as $\mathcal{Q}f = \frac{1}{|G|} \sum_{g \in G} f \circ \rho_g$. If the data distribution is symmetric, then we have that*

$$\mathbb{E}[\|\mathcal{Q}f(x) - s(x)\|_2^2] \leq \mathbb{E}[\|f(x) - s(x)\|_2^2],$$

*for all approximator $f$. Specifically, if $f$ is an output function of GNs, $\mathcal{Q}f$ can be an output function of GENs/LENs with a $|G|$ times model size, implying that GENs/LENs can fix the expressive power gap and achieve a smaller expected loss by a larger model size.*

When the GN has an output function $f(x) = \sum_{i=1}^{n} \beta_i \sigma(\langle \alpha_i, x \rangle)$, the weight matrices of GENs/LENs can be formulated as

$$W^{(2)} = \frac{1}{|G|} [\underbrace{\beta_1, \ldots, \beta_1}_{|G|}, \ldots, \underbrace{\beta_m, \ldots, \beta_m}_{|G|}], \text{ and}$$

$$W^{(1)} = [\rho_{g_1}^T \alpha_1, \ldots, \rho_{g_{|G|}}^T \alpha_1, \ldots, \rho_{g_1}^T \alpha_m, \ldots, \rho_{g_{|G|}}^T \alpha_m]^T,$$

which implies comparable expressive power while requiring a model size proportional to $|G|$. Meanwhile, the weight matrices can be generated in two steps: (1) copy the same number of original channel vectors, and (2) generate all symmetric channel vectors to make the network equivariant. Therefore, although LENs have a larger model size, they have the same number of learnable parameters. As a result, the dimensionality of the hypothesis space can be reduced by selecting fewer original channel vectors or by constraining them to a subspace, while maintaining comparable expressive power.

If we use the method in Lemma 6.1 to compress the networks, we can further get an even better result. More precisely, we can let the weight matrices be

$$W^{(2)} = [\underbrace{\beta_1, \ldots, \beta_1}_{C_1}, \ldots, \underbrace{\beta_m, \ldots, \beta_m}_{C_m}], \text{ and}$$

$$W^{(1)} = [\rho_{g_{11}}^T \alpha_1, \ldots, \rho_{g_{1C_1}}^T \alpha_1, \ldots, \rho_{g_{m1}}^T \alpha_m, \ldots, \rho_{g_{mC_m}}^T \alpha_m]^T,$$

where $\alpha_i$ is constrained to satisfy $|\{\rho_g^T \alpha_i : g \in G\}| = C_i$, and $g_{ij}$ is chosen to satisfy $\{\rho_{g_{i,j}}^T \alpha_i : j \in [C_i]\} = \{\rho_g^T \alpha_i : g \in G\}$. Similarly, this model still has comparable expressive power when targeting invariant target functions, but a lower-dimensional hypothesis space, since we constrain all first-layer channel vectors within a (sub-)space.

### 6.3 Enlarged LEN with Smaller Hypothesis Complexity

While increasing model size allows LENs to achieve comparable expressive power to GNs, a natural concern is that this may harm the generalizability. In this section, we revisit Example 5.1 to demonstrate that despite requiring more neurons, LENs can still maintain a less complex hypothesis space than GNs. This lower complexity is significant as it suggests that LENs can achieve better generalizability.

As discussed in Section 5, a LEN with six neurons and a GN with three neurons can represent the same target function. With the permutation representation $\psi$ defined as

$$\psi_g = \begin{bmatrix} 0 & 1 & & & & \\ 1 & 0 & & & & \\ & & 0 & 1 & & \\ & & 1 & 0 & & \\ & & & & 0 & 1 \\ & & & & 1 & 0 \end{bmatrix}, \psi_h = \begin{bmatrix} & & 1 & 0 & & \\ & & 0 & 1 & & \\ 1 & 0 & & & & \\ 0 & 1 & & & & \\ & & & & 0 & 1 \\ & & & & 1 & 0 \end{bmatrix}.$$

Then the output function of the LEN can be formulated as

$$b_1[\sigma(c_1 x + c_2 y) + \sigma(c_2 x + c_1 y)]$$
$$+ b_1[\sigma(-c_1 x - c_2 y) + \sigma(-c_2 x - c_1 y)]$$
$$+ b_2[\sigma(c_3 x - c_3 y) + \sigma(-c_3 x + c_3 y)].$$

From this formulation, we can determine that is characterized by five parameters, meaning its parameter space is $\mathbb{R}^5$. In contrast, a three-neuron GN is defined as $\sum_{i=1}^{3} \beta_i \sigma(c_1 x + c_2 y)$, which involves 9 parameters, resulting in a larger parameter space of $\mathbb{R}^9$.

The key insight here is that although LENs may have a larger model size to maintain the expressive power, their inherent structural constraints could lead to a lower-dimensional hypothesis space. Consequently, this reduced complexity implies that LENs are expected to exhibit superior generalizability compared to standard GNs in this scenario.

## 7 Conclusions

Equivariant neural networks leverage data symmetries as a structural inductive bias, leading to significant success across various applications. Despite their popularity, their expressive power is not yet fully understood. This study evaluates how enforcing equivariance influences the expressive power of two-layer ReLU networks. By analyzing boundary hyperplanes and channel vectors, we show that these constraints can inherently limit a model's expressive power. We demonstrate, however, that this limitation can be compensated for by increasing model scale, through formal upper bounds for the necessary expansion. Notably, our findings reveal that these larger equivariant architectures possess a reduced hypothesis space dimensionality, suggesting superior generalization potential.

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

## A  Proof of Theorem 4.2

*Proof.* We verify that the boundary hyperplane is well-defined. Firstly, we should check the definition of feature function $\mathcal{F}$. Given the function $F = \sum_{i=1}^{m} \beta_i \sigma(\langle \alpha_i, x \rangle)$, we could consider the set $S = \{ x \in \mathbb{R}^n : \langle \alpha_i, x \rangle \neq 0, \forall i \in [m] \}$. Of course its complementary set $S^c$ is the union of some hyperplanes $\bigcup_{i=1}^{m} M_i$, where $M_i = \{ x : \langle \alpha_i, x \rangle = 0 \}$. Then for any $x \in S$, one of $\langle \alpha_i, x \rangle > 0$ and $\langle \alpha_i, x \rangle < 0$ holds. For small enough $r$, $\langle \alpha_i, x \rangle > 0$ or $\langle \alpha_i, x \rangle < 0$ holds for all $y \in B(x, r)$. As a result, we have $B(x, r) \subset S$. While every $\langle \alpha_i, y \rangle$ does not change its sign for $y \in B(x, r)$, the output function $F$ is linear in $B(x, r)$. Especially by Riesz representation theorem, there exists a unique vector $v$ such that $F(y) = \langle v, y \rangle$ on $B(x, r)$. Combining with that $F(x) = \sum_{i:\langle \alpha_i, x \rangle > 0} \beta_i \langle \alpha_i, x \rangle$, we have $\mathcal{F}(x) = \sum_{i:\langle \alpha_i, x \rangle > 0} \alpha_i \beta_i^T$ for all $x \in S$. Moreover, for every hyperplane $M \neq M_i$ for all $i \in [m]$ and any continuous distribution $m$ on it, we have $x \in S$ almost surely.

For every $x \in M$ of any given hyperplane $M$ and $v \notin M$, we have $\mathcal{F}(x + \delta v)$ is a constant when $\delta > 0$ is small enough, since $x + \delta v$ would not pass through any hyperplanes $M_i$ as $\delta$ goes to 0. As a result, $\mathcal{F}(x + \delta v)$ converges to that constant uniformly, bounded by $\sum_{i=1}^{m} \| \alpha_i \beta_i^T \|$. Therefore, dominated convergence theorem ensures that $\lim \mathbb{E}[\mathcal{F}(x + \delta v)] = \mathbb{E}[\lim \mathcal{F}(x + \delta v)]$. Moreover, if $x + \delta v$ is not in the the hyperplanes $M_i$ (it almost surely holds), $x \in S$ implies $\lim \mathcal{F}(y + \delta v)$ is a constant for all $y \in B(x, r) \cap M$ for a small radius $r$. Since $\lim \mathcal{F}(x + \delta v)$ is piece-wise constant, the expectation is well-defined.

For given $M$, we can compare $\lim \mathcal{F}(x + \delta v)$ and $\lim \mathcal{F}(x - \delta v)$ immediately. If $x \notin M_i$ for all $i \in [m]$, we have $x \in S$ and then $x + \delta v$ and $x - \delta v$ are both in $B(x, r) \subset S$ for small $\delta$, resulting in that $\mathcal{F}(x + \delta v) = \mathcal{F}(x - \delta v)$ for small $\delta > 0$. As a result, if $M \neq M_i$ for all $i \in [m]$, $x$ is in $S$ and therefore $\mathcal{F}(x + \delta v) = \mathcal{F}(x - \delta v)$ almost surely, so that $\lim \mathbb{E}[\mathcal{F}(x + \delta v)] = \lim \mathbb{E}[\mathcal{F}(x - \delta v)]$. Otherwise, we can let $\tilde{F} = F - \sum_{i:\langle \alpha_i, x \rangle = 0, \forall x \in M} \beta_i \sigma(\langle \alpha_i, x \rangle)$ and thus

$$\lim \mathbb{E}[\tilde{\mathcal{F}}(x + \delta v)] = \lim \mathbb{E}[\tilde{\mathcal{F}}(x - \delta v)].$$

Meanwhile, let $G$ denote $F - \tilde{F} = \sum_{i:\langle \alpha_i, x \rangle = 0, \forall x \in M} \beta_i \sigma(\langle \alpha_i, x \rangle)$, we have

$$\mathcal{G}(x + \delta v) - \mathcal{G}(x - \delta v) = \sum_{i:\langle \alpha_i, v \rangle > 0} \alpha_i \beta_i^T - \sum_{i:\langle \alpha_i, v \rangle < 0} \alpha_i \beta_i^T, \forall x \in M.$$

Therefore, we have

$$\mathbb{E}[\mathcal{G}(x + \delta v) - \mathcal{G}(x - \delta v)] = \sum_{i:\langle \alpha_i, v \rangle > 0} \alpha_i \beta_i^T - \sum_{i:\langle \alpha_i, v \rangle < 0} \alpha_i \beta_i^T.$$

As a result, the hyperplane $M$ is boundary hyperplane if and only if $\sum_{i:\langle \alpha_i, v \rangle > 0} \alpha_i \beta_i^T \neq \sum_{i:\langle \alpha_i, v \rangle < 0} \alpha_i \beta_i^T$.

Moreover, if no $\alpha_i$ satisfies that $\langle \alpha_i, M \rangle = 0$, the hyperplane $M$ is not a boundar hyperplane since $\sum_{i:\langle \alpha_i, v \rangle > 0} \alpha_i \beta_i^T = 0 = \sum_{i:\langle \alpha_i, v \rangle < 0} \alpha_i \beta_i^T$. Inversely, if $M$ is a boundary hyperplane, there exists some channel vector $\alpha$ such that $\langle x, M \rangle = 0$.

The proof is completed. In addition, we would like to use the feature function to characterize the boundary hyperplane for future extension to multi-layer conditions, where the equivariant condition would be more complex. However, the feature functions can be similarly defined and therefore the boundary can be shaped.

## B  Proof of Lemma 4.4

For simplicity, we denote $e_i \in \mathbb{R}^n$ as the vector $(0, \dots, 1, \dots, 0)^T$, where the $i$-th entry is 1 and other entries are all 0. Since $\psi_g \circ \sigma = \sigma \circ \psi_g$, we have $\psi \sigma(x) = \sigma(\psi x)$ for all $x \in \mathbb{R}^n$.

If there are two nonzero entries $\psi_{ij} < \psi_{ik}$ in the same row of $\psi_g$, we can set $x = e_j - e_k$ and then we have $(\psi_g \sigma(x))_i = (\psi_g e_j)_i = \psi_{ij}$ while $(\sigma(\psi_g x))_i = \sigma(\psi_{ij} - \psi_{ik}) = 0$. That implies $\psi_{ij} = 0$, which contradicts. Thus, each row has at most one nonzero entry. Hence, there are at most $n$ nonzero entries of $\psi_g$, which are in different rows.

Since $\psi_g$ is invertible, each row and each column of $\psi_g$ have at least one nonzero entry. It implies that there are at least $n$ nonzero entries of $\psi_g$. Thus, there are exactly $n$ nonzero entries of $\psi_g$ and $\psi_g$ is a generalized permutation matrix $\mathrm{diag}(\lambda_g^1, \dots, \lambda_g^m) P_g$.

Moreover, when $\psi_{ij} \neq 0$, we can set $x = e_j$. Then $\sigma(\psi x) = \psi \sigma(x)$ implies that $\sigma(\psi_{ij}) = \psi_{ij}$. From the definition of ReLU, we have that $\psi_{ij} > 0$. The proof is completed. $\qquad\square$

## C   Proof of Theorem 4.3

*Proof.* Let $F$ be an invariant output function that can be represented as $\sum_{i=1}^{m} \beta_i \sigma(\langle \alpha_i, x \rangle)$. Since $F$ is invariant, it can be also reformulated as

$$F = \mathcal{Q}F = \sum_{i=1}^{m} \frac{\beta_i}{|G|} \sum_{g \in G} \sigma(\langle \alpha_i, \rho_g x \rangle).$$

Then the feature function can be

$$\mathcal{F}(x) = \frac{1}{|G|} \sum_{i,g:\langle \rho_g^T \alpha_i, x \rangle > 0} \rho_g^T \alpha_i \beta_i^T.$$

Then, for every transformed point $\rho_g x$, we have

$$\mathcal{F}(\rho_g x) = \frac{1}{|G|} \sum_{i,h:\langle \rho_h^T \alpha_i, \rho_g x \rangle > 0} \rho_h^T \alpha_i \beta_i^T = \frac{1}{|G|} \sum_{i,h:\langle \rho_{hg}^T \alpha_i, x \rangle > 0} (\rho_g^T)^{-1} \rho_{hg}^T \alpha_i \beta_i^T$$

$$= (\rho_g^T)^{-1} \frac{1}{|G|} \sum_{i,h:\langle \rho_h^T \alpha_i, x \rangle > 0} \rho_h^T \alpha_i \beta_i^T = (\rho_g^T)^{-1} \mathcal{F}(x).$$

As a result, $\mathcal{F}$ is somehow "equivariant". Moreover, let us consider the definition of boundary hyperplanes, we have

$$\lim_{\delta \to 0^+} \mathbb{E}[\mathcal{F}(\rho_g x + \rho_g v)] = \lim_{\delta \to 0^+} \mathbb{E}[\mathcal{F}(\rho_g(x + v))] = (\rho_g^T)^{-1} \lim_{\delta \to 0^+} \mathbb{E}[\mathcal{F}(x + v)],$$

and

$$\lim_{\delta \to 0^+} \mathbb{E}[\mathcal{F}(\rho_g x - \rho_g v)] = \lim_{\delta \to 0^+} \mathbb{E}[\mathcal{F}(\rho_g(x - v))] = (\rho_g^T)^{-1} \lim_{\delta \to 0^+} \mathbb{E}[\mathcal{F}(x - v)],$$

where $v \notin M$ and $\rho_g v \notin \rho_g M$. Therefore, $M$ is a boundary hyperplane if and only if $\rho_g M$ is a boundary hyperplane.

Besides, if we consider the equivariant condition, $M$ is a boundary hyperplane if and only if

$$\sum_{\substack{i,h:\langle \rho_h^T \alpha_i, v \rangle > 0 \\ \langle \rho_h^T \alpha_i, M \rangle = 0}} \rho_h^T \alpha_i \beta_i^T \neq \sum_{\substack{i,h:\langle \rho_h^T \alpha_i, v \rangle < 0 \\ \langle \rho_h^T \alpha_i, M \rangle = 0}} \rho_h^T \alpha_i \beta_i^T.$$

By contrast, that for $\rho_g M$ is

$$\sum_{\substack{i,h:\langle \rho_h^T \alpha_i, \rho_g v \rangle > 0 \\ \langle \rho_h^T \alpha_i, \rho_g M \rangle = 0}} \rho_h^T \alpha_i \beta_i^T \neq \sum_{\substack{i,h:\langle \rho_h^T \alpha_i, \rho_g v \rangle < 0 \\ \langle \rho_h^T \alpha_i, \rho_g M \rangle = 0}} \rho_h^T \alpha_i \beta_i^T.$$

The LHS is equal to

$$\sum_{\substack{i,h:\langle \rho_h^T \alpha_i, \rho_g v \rangle > 0 \\ \langle \rho_h^T \alpha_i, \rho_g M \rangle = 0}} \rho_h^T \alpha_i \beta_i^T = \sum_{\substack{i,h:\langle \rho_{hg}^T \alpha_i, v \rangle > 0 \\ \langle \rho_{hg}^T \alpha_i, M \rangle = 0}} (\rho_g^T)^{-1} \rho_{hg}^T \alpha_i \beta_i^T = (\rho_g^T)^{-1} \sum_{\substack{i,h:\langle \rho_h^T \alpha_i, v \rangle > 0 \\ \langle \rho_h^T \alpha_i, M \rangle = 0}} \rho_h^T \alpha_i \beta_i^T,$$

while the RHS is equal to

$$\sum_{\substack{i,h:\langle \rho_h^T \alpha_i, \rho_g v \rangle < 0 \\ \langle \rho_h^T \alpha_i, \rho_g M \rangle = 0}} \rho_h^T \alpha_i \beta_i^T = \sum_{\substack{i,h:\langle \rho_{hg}^T \alpha_i, v \rangle < 0 \\ \langle \rho_{hg}^T \alpha_i, M \rangle = 0}} (\rho_g^T)^{-1} \rho_{hg}^T \alpha_i \beta_i^T = (\rho_g^T)^{-1} \sum_{\substack{i,h:\langle \rho_h^T \alpha_i, v \rangle < 0 \\ \langle \rho_h^T \alpha_i, M \rangle = 0}} \rho_h^T \alpha_i \beta_i^T.$$

As a result, $M$ and $\rho_g M$ have the equivariant conditions to be boundary hyperplanes, causing that $M$ is a boundary hyperplane if and only if $\rho_g M$ is a boundary hyperplane. The proof is completed. $\qquad\square$

## D   Proof of Theorem 4.6

*Proof.* Due to $W^{(1)} \circ \rho_g = \psi_g \circ W^{(1)}$, and thus we have

$$W^{(1)} = \frac{1}{|G|} \sum_{g \in G} \psi_g^{-1} \circ W^{(1)} \circ \rho_g.$$

Then $F$ can be rewritten as:

$$F(x) = [\beta_1 \ldots \beta_m] \, \sigma \left( [\alpha_1 \quad \cdots \quad \alpha_m]^\top x \right)$$

$$= [\beta_1 \ldots \beta_m] \, \sigma \left( \frac{1}{|G|} \sum_{g \in G} \left( \psi_g^{-1} [\alpha_1 \quad \cdots \quad \alpha_m] \right)^\top \rho_g x \right)$$

$$= [\beta_1 \ldots \beta_m] \, \sigma \left( \frac{1}{|G|} \sum_{g \in G} \left[ \frac{\alpha_{P_g(1)}}{(\psi_g)_{1,P_g(1)}} \quad \cdots \quad \frac{\alpha_{P_g(m)}}{(\psi_g)_{m,P_g(m)}} \right]^\top \rho_g x \right)$$

$$= \frac{1}{|G|} \sum_{i=1}^m \beta_i \sigma \left( \sum_{g \in G} \left( \frac{\alpha_{P_g(i)}}{(\psi_g)_{i,P_g(i)}} \right)^\top \rho_g x \right)$$

$$= \frac{1}{|G|} \sum_{i=1}^m \frac{\beta_i}{\|\beta_i\|} \sigma \left( \sum_{g \in G} \left( \frac{\|\beta_i\|}{(\psi_g)_{i,P_g(i)}} \alpha_{P_g(i)} \right)^\top \rho_g x \right)$$

$$= \sum_{i=1}^m \frac{\beta_i}{\|\beta_i\|} \sigma \left( \left( \sum_{g \in G} \frac{1}{|G|} \rho_g^\top \|\beta_{P_g(i)}\| \alpha_{P_g(i)} \right)^\top x \right).$$

Denote $\tilde{\beta}_i = \frac{\beta_i}{\|\beta_i\|}$ and $\tilde{\alpha}_i = \sum_{g \in G} \frac{1}{|G|} \rho_g^\top \|\beta_{P_g(i)}\| \alpha_{P_g(i)}$, we get $F(x) = \sum_{i=1}^m \tilde{\beta}_i \sigma \langle \tilde{\alpha}_i, x \rangle$.

For new parameters, we have

$$\rho_h^\top \tilde{\alpha}_i = \rho_h^\top \sum_{g \in G} \frac{1}{|G|} \rho_g^\top \beta_{P_g(i)} \alpha_{P_g(i)}$$

$$= \sum_{g \in G} \frac{1}{|G|} \rho_{gh}^\top \beta_{P_g(i)} \alpha_{P_g(i)}$$

$$= \sum_{g \in G} \frac{1}{|G|} \rho_{gh}^\top \beta_{P_{gh}(P_{h^{-1}}(i))} \alpha_{P_g(P_{h^{-1}}(i))}$$

$$= \tilde{\alpha}_{P_{h^{-1}}(i)},$$

which is a satisfied rewritten LEN.   $\square$

## E   Proof of ENs' Universal Represetation Ability

*Proof.* We construct the following ENs to represent all equivariant output functions of GNs. Let the output function of GNs be $W^{(L)} \sigma(W^{(L-1)} \sigma(\ldots \sigma(W^{(1)} x) \ldots))$. Let the weight matrices of ENs be

$$\tilde{W}^{(L)} = \frac{1}{|G|} (\underbrace{\phi_{g_1}^{-1} W^{(L)}, \ldots, \phi_{g_{|G|}}^{-1} W^{(L)}}_{|G|})$$

$$\tilde{W}^{(\ell)} = \text{diag}(\underbrace{W^{(\ell)}, \ldots, W^{(\ell)}}_{|G|}), \forall \ell = 2, 3, \ldots, L-1$$

$$\tilde{W}^{(1)} = (\underbrace{(W^{(1)} \rho_{g_1})^T, \ldots, (W^{(1)} \rho_{g_{|G|}})^T}_{|G|})^T,$$

where the group representation $\psi$ in the intermediate layers is the corresponding permutation representation acting on each axis $e_{g_i}$ as $\psi_g e_{g_i} = e_{g_i g^{-1}}$. As a result, we have, $\forall \ell = 2, 3 \ldots, L-1$,

$$
\tilde{W}^{(1)} \rho_g = \begin{bmatrix} W^{(1)} \rho_{g_1} \\ W^{(1)} \rho_{g_2} \\ \vdots \\ W^{(1)} \rho_{g_{|G|}} \end{bmatrix} \rho_g = \begin{bmatrix} W^{(1)} \rho_{g_1 g} \\ W^{(1)} \rho_{g_2 g} \\ \vdots \\ W^{(1)} \rho_{g_{|G|} g} \end{bmatrix} = \psi_g \begin{bmatrix} W^{(1)} \rho_{g_1} \\ W^{(1)} \rho_{g_2} \\ \vdots \\ W^{(1)} \rho_{g_{|G|}} \end{bmatrix} = \psi_g \tilde{W}^{(1)},
$$

$$
\tilde{W}^{(\ell)} \psi_g = \mathrm{diag}(\underbrace{W^{(\ell)}, \ldots, W^{(\ell)}}_{|G|}) \psi_g = \mathrm{diag}(\underbrace{W^{(\ell)}, \ldots, W^{(\ell)}}_{|G|}) = \psi_g \mathrm{diag}(\underbrace{W^{(\ell)}, \ldots, W^{(\ell)}}_{|G|}) = \psi_g \tilde{W}^{(\ell)},
$$

$$
\tilde{W}^{(L)} \psi_g = \frac{1}{|G|} (\underbrace{\phi_{g_1}^{-1} W^{(L)}, \ldots, \phi_{g_{|G|}}^{-1} W^{(L)}}_{|G|}) \psi_g = \frac{1}{|G|} (\underbrace{\phi_{g_1 g^{-1}}^{-1} W^{(L)}, \ldots, \phi_{g_{|G|} g^{-1}}^{-1} W^{(L)}}_{|G|})
$$

$$
= \frac{1}{|G|} \phi_g (\underbrace{\phi_{g_1}^{-1} W^{(L)}, \ldots, \phi_{g_{|G|}}^{-1} W^{(L)}}_{|G|})
$$

As a result, the constructed network has the same output function $\mathcal{Q}F$ and equivariant layers, renders it an EN. The proof is completed. $\square$

## F   Proof of Lemma 6.1

*Proof.* Denote $M = \{\rho_g^T \alpha / \|\rho_g^T \alpha\| : g \in G\}$ and $N = \{\rho_g^T \beta / \|\rho_g^T \beta\| : g \in G\}$, where $\alpha$ and $\beta$ are two nonzero vectors in $\mathbb{R}^n$. Then we prove that $M \cap N = \emptyset$ or $M = N$.

If $M \cap N \neq \emptyset$, there exist group elements $g_0, h_0 \in G$ such that $\rho_{g_0}^T \alpha / \|\rho_{g_0}^T \alpha\| = \rho_{h_0}^T \beta / \|\rho_{h_0}^T \beta\|$. Then, for any element $\rho_g^T \alpha / \|\rho_g^T \alpha\| \in M$, there exists $\rho_{h_0 g_0^{-1} g}^T \beta / \|\rho_{h_0 g_0^{-1} g}^T \beta\| \in N$ such that

$$
\rho_{h_0 g_0^{-1} g}^T \beta = \rho_g^T \rho_{g_0^{-1}}^T \rho_{h_0}^T \beta = \frac{\|\rho_{h_0}^T \beta\|}{\|\rho_{g_0}^T \alpha\|} \cdot \rho_g^T \alpha \quad \text{and} \quad \frac{\rho_{h_0 g_0^{-1} g}^T \beta}{\|\rho_{h_0 g_0^{-1} g}^T \beta\|} = \frac{\rho_g^T \alpha}{\|\rho_g^T \alpha\|}.
$$

It implies that $M \subset N$. Inversely, we can prove that $N \subset M$ in the same way. Hence, combining $M \subset N$ and $N \subset M$, we have $M = N$. Moreover, when denoting $\widetilde{M} = \{\rho_g^T \alpha : g \in G\}$, we can also prove that $|\widetilde{M}| = |M|$. If there exist $g, h \in G$ such that $\rho_g^T \alpha / \|\rho_g^T \alpha\| = \rho_h^T \alpha / \|\rho_h^T \alpha\|$, we have $\rho_g^T \alpha = c \rho_h^T \alpha$ with some positive number $c$ since they have the same direction. Then, we have $\rho_{gh^{-1}}^T \alpha = c\alpha$ and

$$
(\rho_{gh^{-1}}^T)^{|G|} \alpha = (\rho_{gh^{-1}}^T)^{|G|-1} c\alpha = (\rho_{gh^{-1}}^T)^{|G|-2} c^2 \alpha = \cdots = c^{|G|} \alpha.
$$

Meanwhile, since $(gh^{-1})^{|G|} = e \in G$, we have $(\rho_{gh^{-1}}^T)^{|G|} \alpha = \rho_e^T \alpha = \alpha$. It implies that $c^{|G|} = 1$ and thus $c = 1$. Finally, we have that $\rho_g^T \alpha = \rho_h^T \alpha \Leftrightarrow \frac{\rho_g^T \alpha}{\|\rho_g^T \alpha\|} = \frac{\rho_h^T \alpha}{\|\rho_h^T \alpha\|}$, which indicates that $|\widetilde{M}| = |M|$. Therefore, if $M = N$, we have $\widetilde{M} = k\widetilde{N}$ for some $k > 0$.

In particular, if there exist two parallel channel vectors $\rho_s^T \alpha_i$ and $\rho_t^T \alpha_j$, we can assume $\rho_s^T \alpha_i = k \rho_t^T \alpha_j$ for some $k > 0$ and then we have

$$
\mathcal{O}_i = \{\rho_g^T (\rho_s^T \alpha_i) : g \in G\} = \{\rho_g^T (k \rho_t^T \alpha_j) : g \in G\} = k\{\rho_g^T (\rho_t^T \alpha_j) : g \in G\} = k\mathcal{O}_j.
$$

Moreover, ENs can merge these channels as

$$
\sum_{g \in G} \beta_i \sigma(\langle \rho_g^T \alpha_i, x \rangle) + \beta_j \sigma(\langle \rho_g^T \alpha_j, x \rangle) = \sum_{g \in G} (k\beta_i + \beta_j) \sigma(\langle \rho_g^T \alpha_j, x \rangle)
$$

As a result, we can find an EN of the same output function such that $\rho_g^T \alpha_i / \|\rho_g^T \alpha_i\| \neq \rho_h^T \alpha_j / \|\rho_h^T \alpha_j\|$ for all $i \neq j$.

Furthermore, we can define the stabilizer subgroup $Stab(\alpha) = \{g : \rho_g^T \alpha_i\}$ and therefore $G = Stab(\alpha) \oplus \tilde{M}$ with

$$\sum_{g \in G} f(\rho_g^T \alpha) = \sum_{h \in G/\text{Stab}(\alpha)} \sum_{gh^{-1} \in \text{Stab}(\alpha)} f(\rho_g^T \alpha) = |\text{Stab}(\alpha)| \cdot \sum_{h \in G/\text{Stab}(\alpha)} f(\rho_h^T \alpha).$$

The proof is completed. □

## G   Proof of Theorem 6.2

*Proof.* We consider the following conditions with given invariant output function of GNs $F(x) = \sum_{i=1}^m \beta_i \sigma(\langle \alpha_i, x \rangle)$. Denote the number of boundary hyperplane of $F(x)$ by $N$.

If $N \leq m - 1$, we can construct the EN of Lemma 6.1, where we let the output function be formulated as $\mathcal{Q}F(x) = \frac{1}{|G|} \sum_{i=1}^m \beta_i \sum_{g \in G} \sigma(\langle \rho_g^T \alpha_i, x \rangle)$. Lemma 6.1 implies that the number of channel vectors $m'$ after merging is at most double the number of boundary hyperplanes plus 2, and therefore,

$$m' \leq 2N + 2 \leq 2(m - 1) + 2 = 2m.$$

If $N = m$, there are $m$ boundary hyperplanes $M_1, \ldots, M_m$ and we can assume $\langle \alpha_i, M_i \rangle = 0$ for all $i \in [m]$ without loss of generality. Theorem 4.3 indicates that all the boundary hyperplanes are symmetric, enable us to classifier them as $S_1 = \{M_{k_0+1}, \ldots, M_{k_1}\}, \ldots, S_{l+1} = \{M_{k_l+1}, \ldots, M_{k_{l+1}}\}$ with $k_0 = 0$ and $k_{l+1} = m$, where each set collects all symmetric boundary hyperplanes. Then consider the orbits $\mathcal{O}_i = \{\rho_g^T \alpha_i : g \in G\}$, we have $\mathcal{O}_i = k_{ij} \mathcal{O}_j$ for all $i, j$ in the same classification set $S$. Lemma 6.1 implies that the sum $\frac{1}{|G|} \sum_{i=1}^m \beta_i \sum_{g \in G} \sigma(\langle \rho_g^T \alpha_i, x \rangle)$ can be formulated by the sum of at most $2m$ items, resulting in $m' \leq 2m$ also.

The proof is completed. □

