# OpenReview forum: "Drawback of Enforcing Equivariance and its Compensation via the Lens of Expressive Power"
_TMLR — Accepted by TMLR_

### Review · Reviewer_AX99 · 2026-02-14

**Summary Of Contributions:**

This paper draws attention to shortcomings of equivariant neural networks. The main claims is that there is a trade off for restricting the
hypothesis space to equivariant models increases the expected risk when approximating targets. To address this, the authors compared risks for GN, GEN, and LEN models and also showed that the symmetry requirements incurred larger amount to neurons to have similar expressive power with unconstrained models. I think novelty of their position and theoretical rigor are noticeable strength of this work.

**Audience:**

Yes

**Audience Explanation:**

Equivalent neural nets are an active field of study in machine learning and its applications.

**Claims And Evidence:**

Yes

**Claims Explanation:**

This is a theoretical work evidenced by proofs. I did not checked the entire steps, but I think most of the arguments are sound at least in the outlined example. Most of its claims can be intuitively understood by ML practitioners.

However, I am currently somewhat doubtful for implication of their claim and whether their claim holds in practice.
* I believe the entire proofs are based on all models consider the identical risk minimization problem i.e. $\inf_\theta \mathbb{E} p \lVert F_\theta(x) - x\rVert^2$. However, there are plenty of cases that equivariant models and nonequivariant models cannot be directly compared with same cost functions and data distributions can be drawn from manifolds equipped with symmetry that can be only trained with equivariant models.
* In the same vein, I think the authors arguments are trivial in the sense that nonequivariant models are **specialized** to solve **unconstrained** risk minimization while equivariant models are not. Therefore, I believe that it would be much meaningful if the analysis focused on highlighting the trade-off of enforcing equivariance, rather than one-sided drawback of equivalent nets, where it is apparent that the general cost minimization is was not likely a preferred objective at the fist place.
* It will be much appreciated if the analysis contain extension to 3D spaces e.g. rotation group.

**Requested Changes:**

* Please put appropriate justification for comparing expressive power with unconstrained risks.
* Some practical group convolution manages to cut their computation costs with efficient modeling that can be only possible with equivariance. I suggest the authors to extend the conclusion section with limitations to properly represent their theory in the real-world.
* Please hints how these theoretical findings in 2D can be applied to  group equivariant 3D  deep learning models such as SO(3).

---

### Review · Reviewer_RR2G · 2026-02-24

**Summary Of Contributions:**

The paper discusses equivariant two-layer ReLU-networks. In particular, the discussion centers on the fact that restricting each linear layer to be equivariant under a permutation representation does not cover all possible equivariant functions representable by the base architecture at a fixed width. The paper notes that one can compensate for this by increasing the network width.

**Audience:**

No

**Audience Explanation:**

The findings in the paper are interesting to the geometric deep learning community, but the presentation of the findings should be improved. Most of the findings are already covered in the literature. The relevant papers should be discussed and compared to. It needs to be clear what the presented findings contribute that has not been priorly covered.

1. Most importantly, in [A] all invariant shallow ReLU-networks are classified and it is shown that they can always be rewritten such that the layers are equivariant under signed permutation representations. Furthermore, these signed permutation representations can be written as permutation representations by doubling the network width. Theorem 6.2 in the submission therefore seems to follow directly from the results in [A]. The same is true for Section 5.
2. Lemma 4.4 is well-known, see for instance [B]. A similar proof to the one in the submission appears in [C].
3. Theorem 6.3 is covered in [D].

[A] Agrawal & Ostrowski, A Classification of G-Invariant Shallow Neural
Networks, NeurIPS 2022

[B] Godfrey et al, On the Symmetries of Deep Learning Models and their Internal Representations, NeurIPS 2022

[C] Bökman & Kahl, Investigating how ReLU-networks encode symmetries, NeurIPS 2023

[D] Elesedy & Zaidi, Provably Strict Generalisation Benefit for Equivariant Models, ICML 2021

**Claims And Evidence:**

Yes

**Claims Explanation:**

There are some slight inaccuracies, but they seem to be easily fixable.

1. Some results are stated in terms of invarance only, it would help the reader to be clearer about which results regard invariance vs equivariance.
2. The last paragraph on page 5 is unclear. It is stated that "*We prove that for a GEN, the set of its boundary hyperplanes are symmetric with respect to all group representations.*" However, I believe that the intent is that the symmetry is w.r.t. the specific group representation $\rho$ that the GEN is equivariant under.
3. The first equation in page 6 is stated in terms of invariance. It seems to me like a $\phi$ should be involved for this to cover the equivariant case. Specifically, $\mathcal F(\rho x) = \rho^{-T} \mathcal F(x) \phi^T$.
4. At the bottom of page 10, the following sentence is unclear to me. "*Similarly, this model still has comparable expressive power, but a lower-dimensional hypothesis space, since we constrain all first-layer channel vectors within a (sub-)space.*" Having comparable expressive power seems contradictory to having a smaller hypothesis space. Is the intended meaning that the model has comparable expressive power amongst equivariant functions, but a smaller hypothesis space when considering all functions?

**Requested Changes:**

For me to recommend acceptance of this submission, the points listed under **evidence** above should be clarified and the presented findings should be contextualized within the literature.

---

### Review · Reviewer_CP2c · 2026-03-06

**Summary Of Contributions:**

The paper studies differences in expressive power of General Equivariant Nets (GENs) and Layer-wise Equivariant Nets (LENs), compared to non-equivariant General Networks (GNs), where all the networks are assumed to be 2-layer ReLU networks. They show that in some cases, GENs and LENs clearly lead to higher error than GNs, and that LENs need to have more neurons to approximate functions that GENs can for the same error value. By increasing the number of neurons in LENs to at most 2x the number of neurons in GENs, the expressive capacity of the LEN becomes comparable to the GEN.

Strengths
1. As the paper is fully theoretical, it is easy to see that it is technically sound.
2. The paper addresses some interesting questions about expressivity of equivariant networks.
3. I appreciate the authors constructing simple, but very illustrative examples, to help understand the points about loss of expressivity with equivariance, and how LENs can have the same capacity as GENs with at most 2x the neurons, and still have better generalization.

Weaknesses:
1. As a researcher who does not do a lot of theoretical work, it is not clear to me how useful these results are in practice. All the analysis is for finite groups and for 2-layer ReLU networks. The results mostly seem to agree with general intuitions, which is good, and it is also good to have actual proofs rather than just intuitions, but I don't see broader use of the results. It would be nice if the authors can provide some explanations to put the paper in broader context.

2. The authors should try and improve the Related Work section. Under "Theoretical Analysis", many papers are listed, but no connections to this paper are mentioned, how are all these previous papers related to this submission? The authors may find this paper to be of value to discuss as well, which also considers symmetries in data and misspecified symmetries: https://arxiv.org/abs/2305.17592

3. No proof is provided for Theorem 6.3, I can see that it is true, still good to provide a full proof. I think this paper has some similar ideas: https://arxiv.org/abs/1804.10306

4. The analysis uses number of neurons as the way to measure the size of the networks. This is not really what people think of when they think of sizes of network, the number of learnable parameters that matters more. This is obvious when weight sharing is involved, as in most equivariant networks (like CNNs). So, counting the number of neurons in LENs, and then saying that LEN is still more generalizable because it actually has fewer learnable parameters seems trivial to me (section 6.3). Hope the authors can clarify why the number of neurons is important to count, rather than the learnable parameters.

5. This is not a real weakness, but it would be good if the authors could have a small addition to discuss the impact of training data and the training dynamics on choice of model architectures, as those are not considered in the paper.

6. There seem to be a few typos in the paper. For example, at the beginning of section 3, I think the dimensions of W1 and W2 should be transposed for the dimensions to make sense.

**Audience:**

Yes

**Audience Explanation:**

Yes, TMLR audience includes researchers working on equivariant neural networks. They are likely to find this paper interesting.

**Broader Impact Concerns:**

No such concerns.

**Claims And Evidence:**

Yes

**Claims Explanation:**

All the claims in the paper are mathematical in nature, and they come with mathematical theorems. I have gone through the proofs of all the theorems and lemmas, and they all appear to be correct.

**Requested Changes:**

Please address the weaknesses listed above.

---

### Decision · Action_Editor_RtGK · 2026-04-13

**Recommendation:** Accept as is

**Additional Comments:**

Even though I'm suggesting to accept the paper as is, I would encourage the authors to still clarify and discuss the counting neurons as a proxy for the expressivity question raised by reviewer CP2c. While I think author's give a reasonable response, I still wonder how the weight sharing would affect this. I think this could be just very briefly addressed in the Section 3.

**Audience:**

Yes

**Audience Explanation:**

All the reviewers found the insights of the paper interesting for the TMLR audience. The key results reviewers highlighted were the simple examples to illustrate the requirement of larger amount of parameters to meet the equality constraint, as well as the theoretical results that show how the number of parameters need to be scaled to close the expressivity gap.

**Claims And Evidence:**

Yes

**Claims Explanation:**

This is a fully theoretical work, and all the reviewers found the proofs to be clear and convincing. Reviewer RR2G pointed out some missing prior work, which authors have now added in the revised version. All the reviewers were satisfied with the answers author's had for the concerns regarding accuracy and clarity.

---

> ### Author Response · Authors · 2026-04-18
> **Thanks! All addressed. Finanl copy uploaded**
>
> Thanks very much for your recommendation as well as the additional comments - all have been duly addressed. The camera-ready version is also uploaded.